OBSERVATION

# Self-Collected Gargle Lavage Allows Reliable Detection of SARS-CoV-2 in an Outpatient Setting

Johannes Zander,[a,b] Stephan Scholtes,[c] Maximilian Ottinger,[d] Marcel Kremer,[a] Azadeh Kharazi,[a] Vanessa Stadler,[a] Julia Bickmann,[a] Christian Zeleny,[c] Johannes W. P. Kuiper,[e] ◉Christof R. Hauck[e,f]

[a]Labor Dr. Brunner, Konstanz, Germany
[b]Institut für Laboratoriumsmedizin, Klinikum der Universität München, Munich, Germany
[c]Praxis Dr. S. Scholtes/C. Zeleny, Konstanz, Germany
[d]Praxis Maximilian Ottinger, Konstanz, Germany
[e]Lehrstuhl Zellbiologie, Universität Konstanz, Konstanz, Germany
[f]Konstanz Research School Chemical Biology, Universität Konstanz, Konstanz, Germany

Johannes Zander and Stephan Scholtes contributed equally to this work. The author order was determined by rolling dice.

**ABSTRACT** Current procurement of specimens for severe acute respiratory syndrome coronavirus 2 (SARS-CoV-2) detection requires trained personnel and dedicated equipment. We compared standard nasopharyngeal swabs with self-collected gargle lavage fluid obtained from 80 mostly symptomatic outpatients. After RNA extraction, RT-PCR to detect SARS-CoV-2 was performed. Qualitative results obtained with the paired samples from individual outpatients were 100% congruent. Therefore, self-collected gargle lavage fluid can serve as a suitable specimen for coronavirus disease 2019 (COVID-19) testing in outpatients.

**IMPORTANCE** The SARS-CoV-2 pandemic still strains health care systems worldwide. While COVID-19 testing is considered an essential pillar in combating this infectious disease, shortages in supplies and trained health care personnel often limit the procurement of patient samples, in particular in outpatient settings. Here, we compared the simple self-collection of gargle lavage fluid with the gold standard nasopharyngeal swab as a specimen for COVID-19 testing. By finding complete congruence of results obtained with paired samples of a sizeable patient cohort, our results strongly support the idea that the painless self-collection of gargle lavage fluid provides a suitable and uncomplicated sample for reliable SARS-CoV-2 detection.

**KEYWORDS** COVID-19, RT-PCR, SARS-CoV-2, gargle lavage, outpatients

Fast and reliable testing of persons with suspected severe acute respiratory syndrome coronavirus 2 (SARS-CoV-2) infection is a key element in combating the coronavirus disease 2019 (COVID-19) pandemic. Early in infection, high viral loads can be found in the nasal and oral cavities. Hence, nasopharyngeal swab specimens (NPS) are considered the gold standard material for SARS-CoV-2 testing in this period. However, obtaining NPS by medical staff has several drawbacks: the shortage of trained professionals, significant discomfort for patients, and an increased risk of infection for the medical staff during the procedure. In outpatient settings, these factors can compromise sampling and can delay diagnosis of patients with COVID-19. To achieve maximum testing throughput, alternative sampling strategies, such as self-collection of saliva or gargle lavage (GL) fluid, have been suggested (1–6). However, saliva is usually viscous and inhomogeneous, with mostly low volume, thus hampering the optimized high-throughput workflow in clinical laboratories. In contrast, GL fluid is not subject to these constraints and can be easily self-collected by patients. Recent studies with

Address correspondence to Christof R. Hauck, christof.hauck@uni-konstanz.de.

SARS-CoV and SARS-CoV-2 have indicated that self-collected GL samples could represent a suitable replacement for NPS (1–5, 7–12) for both rather early COVID-19 stages and advanced stages. However, the number of patients tested in some studies is rather small (≤5) (1, 3, 5), GL samples have not been paired with NPS (2, 8), and other studies exclusively focused on material from hospitalized patients with advanced COVID-19 (1, 4, 5, 7, 8). In other cases, GL material was obtained from patients only after they tested positive by NPS, or the GL material was stored until the corresponding NPS result was available, prohibiting a direct comparison of paired samples from the same patient in the same PCR run (10–12).

We compared the suitability of self-collected GL versus NPS taken by health care professionals as testing materials for reverse-transcription PCR (RT-PCR)-based SARS-CoV-2 detection in outpatients.

First, we asked if a GL specimen is suitable for procuring cellular material to an extent comparable to that of the standard NPS, which samples mucosal material in a locally defined area. To this end, we obtained paired GL specimens and NPS from 11 healthy volunteers, isolated RNA, and performed RT-PCR with primers directed against the human RNase P gene as a housekeeping gene. Interestingly, GL fluid yielded similar but slightly larger amounts of host material (Fig. 1A). This finding suggests that the higher dilution of the gargle sample (a 5-ml volume versus a 1-ml volume of the swab sample) is more than compensated for by the larger surface area reached by the gargle lavage. Accordingly, both procedures are suited to extract similar amounts of primary sample.

Next, we recruited patients with possible/probable SARS-CoV-2 infection between October and December 2020. Participants had been traced by health authorities as close contacts of SARS-CoV-2-positive persons and had attended different doctors' offices. The sampling took place when patients first visited doctors' offices because of possible COVID-19. The study protocol (DRKS number DRKS00023904) was approved by the Institutional Review Board of the University of Konstanz, Germany, and was carried out according to the principles of the Declaration of Helsinki. Written informed consent was obtained from all participants. From each patient, a nasopharyngeal swab (Copan eSwabs [Copan; MAST Group] with 1 ml Amies preservation medium [APM]) (13) was taken by professional medical staff. Directly before or after this procedure, a GL sample (30 s gargling of 5 ml 0.9% NaCl) was self-collected by the patients. Samples were sent to the laboratory (Labor Dr. Brunner, Konstanz, Germany), where RT-PCR testing was performed on each of the submitted materials from each patient in parallel. For RNA isolation, 200 $\mu$l of the NPS (stirred in 1 ml APM) or 200 $\mu$l of the GL specimen was used. From these samples, RNA was isolated (QIAcube HT; Qiagen), and RT-PCR was performed using the Rida Gene SARS-CoV-2 assay (R-Biopharm) (14) according to the manufacturer's instructions. Both samples for each patient were analyzed in the same RT-PCR run.

Of 80 patients enrolled in the study, 26 (32.5%) tested positive for SARS-CoV-2 using the professionally acquired NPS (referred to here as positive patients). The gender and age distribution and the reported symptoms of all patients are summarized in Table 1. Importantly, for all of the 26 positive patients, the self-collected material (GL fluid) also produced positive test results (Fig. 1B and Table 1). Moreover, all persons who tested negative for SARS-CoV-2 with the NPS (54 of 80 patients; 67.5%) were consistently negative using GL (Table 1).

Upon direct comparison of the RT-PCR cycle threshold ($Cq$) values of the 26 positive patients, a substantial difference was observed between the self-collected samples (median $Cq$ value for GL fluid of 26.8 [17.2 to 34.6]) and the NPS (median $Cq$ value, 19.4 [13.5 to 34.6]; $P < 0.001$) (Fig. 1B), corresponding to higher virus concentrations in NPS taken by professional health care staff. Overall, only a small portion ($n = 4$) of the positive samples showed a low viral titer with $Cq$ values of $>30$, in line with the idea that participants were identified early during the course of infection around symptom onset, when viral titers are highest (15). Interestingly, when we stratified the NPS samples into a lower half ($Cq$ values below the median of 19.4) and a higher half ($Cq$ values

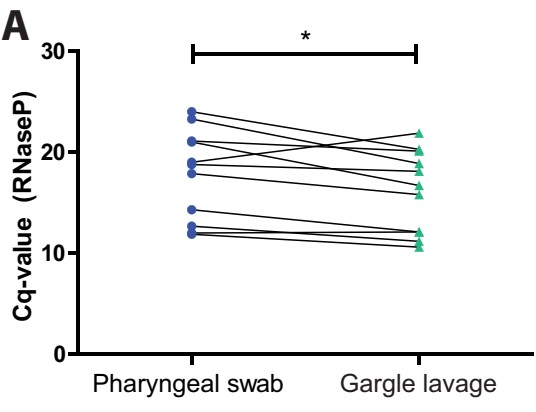

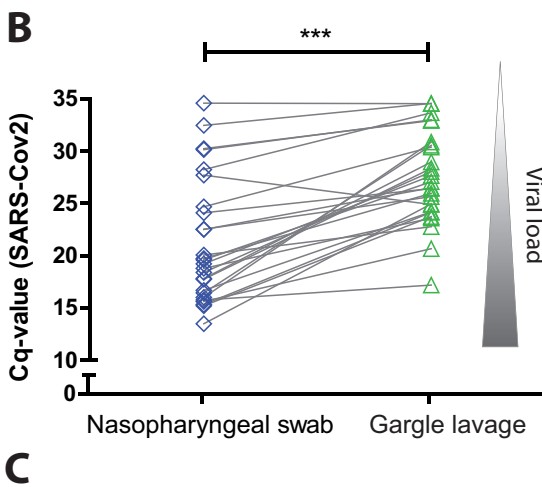

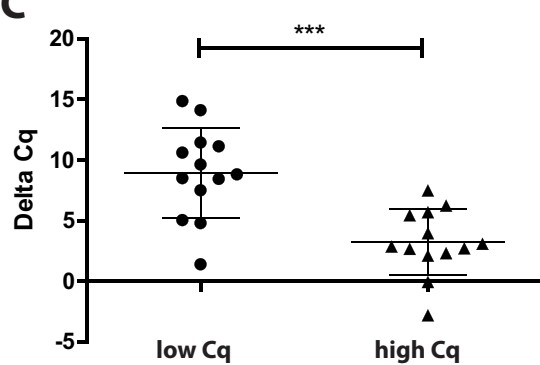

**FIG 1** Cycle threshold (*Cq*) values of different specimens tested for SARS-CoV-2 by RT-PCR. (A) Volunteers (*n* = 11) took (nonprofessionally) pharyngeal swabs (Copan eSwab) and then dipped those into the accompanying tubes each containing 1 ml of Amies preservation medium. Gargle lavage was performed by gargling with 5 ml of 0.9% saline for 20 s. RNA was extracted from 200 $\mu$l of each sample (QIAamp Viral RNA kit). RNA (5 $\mu$l) was added to each RT-PCR mixture (Vulcano3G; myPOLS), and RNase P was detected by TaqMan (CDC-recommended primers/probe). NPS yielded similar but slightly higher *Cq* values for RNase P than GL (Shapiro-Wilk analysis for normal distribution, *P* > 0.1; paired Student's *t* test, *P* < 0.05). (B) *Cq* values of the paired NPS and GL specimens from the 26 outpatients which tested positive for SARS-CoV-2. Nasopharyngeal swabs (Copan eSwab with 1 ml preservation medium) were taken by professional medical staff, and gargle lavage samples (5 ml 0.9% saline) were obtained by the patients themselves. After RNA isolation, paired samples (nasopharyngeal swabs and gargle lavage fluid) were evaluated in parallel by RT-PCR. For more details, see the text. Significance was analyzed by Wilcoxon signed rank test. ***, *P* < 0.001. (C) $\Delta Cq$ values were calculated for each NLS/GL pair by subtracting the respective *Cq* values (GL *Cq* − NLS *Cq*). $\Delta Cq$ values were split in two equal groups based on the *Cq* value of the NLS sample (high *Cq* and low *Cq*; *n* = 13 each). $\Delta Cq$ values were significantly higher in the low- than in the high-*Cq* group (median, 8.8 versus 2.9) (unpaired Student's *t* test). ***, *P* < 0.001.

**TABLE 1** Age and gender distribution of the patient cohort, test results, and reported symptoms

| Characteristic | Value for patient group | |
| --- | --- | --- |
| | RT-PCR positive | RT-PCR negative |
| No. with: | | |
| Professionally sampled NPS | 26 | 54 |
| Self-collected gargle lavage fluid | 26 | 54 |
| Male/female [no. (%)] | 11 (42)/15 (58) | 21 (39)/33 (61) |
| Age (yrs) [range (median)] | 18–89 (33) | 13–77 (30) |
| % with symptom[a] | | |
| Cough | 73 | 68.5 |
| Coryza | 73 | 68.5 |
| Other respiratory symptoms | 53.8 | 31.5 |
| Elevated temperature | 46.2 | 38.9 |
| Loss of smell/taste | 38.5 | 24.1 |

[a]As reported by the study patients.

equal to or above 19.4), we observed that the differences in $Cq$ values between NPS and GL samples were substantially more pronounced for samples which showed lower $Cq$ values in NPS (mean difference in $Cq$ values between NPS and GL, 8.8 for $Cq$ NPS values of <19.4; $n = 13$) (Fig. 1C). In contrast, samples with higher $Cq$ values in NPS ($Cq$ NPS values $\geq$ 19.4; $n = 13$) had a reduced difference (mean difference of 2.9) in $Cq$ values between NPS and GL fluid (Fig. 1C). Moreover, the 4 persons with $Cq$ values in NPS of > 30 showed an even smaller mean difference in $Cq$ values between NPS and GL (mean difference of 1.9).

These findings demonstrate that the detection of SARS-CoV-2 from the (mostly symptomatic) participants by self-collected specimens yields congruent qualitative results. Accordingly, self-collected GL specimens may be suitable for detection of SARS-CoV-2 in symptomatic outpatients by RT-PCR. Furthermore, we showed that viral RNA levels were significantly higher in NPS than in GL and this difference was particularly pronounced at high virus concentrations, whereas the difference between NPS and GL was small at lower virus concentrations. One possible explanation for this might be the altered tissue distribution of the virus during the course of the COVID-19 infection. The early phase of the disease is characterized by high viral loads in the upper airways (15). Here, the virus might be most efficiently isolated by NPS, resulting in lower $Cq$ values compared to GL specimens. However, also at this early phase there is enough virus contained in GL to reliably detect SARS-CoV-2. At later stages, when viral concentrations are generally lower, the virus has moved further down the airways and might be as efficiently collected by GL as by NPS. Consequently, the testing from GL fluid may be sufficiently sensitive to detect SARS-CoV-2 infection at both early and later stages of the infection. This idea is in line with the findings by Mittal et al. (4), who found only marginal differences between GL fluid and NPS for inpatients with progressed COVID-19. Further studies with a longitudinal comparison of different sampling procedures from the onset of symptoms are needed to substantiate this hypothesis. Most importantly, our results with a cohort of outpatients strongly suggest that GL fluid is a valid sampling material during early stages of COVID-19. Even though higher numbers of viral particles can be procured by NPS during this initial period, GLs suffice for reliable detection of SARS-CoV-2 by RT-PCR.

We believe that these results are important, since the numbers of outpatients substantially exceed those of inpatients, and this self-collection technique saves human and material resources and helps to protect medical professionals from infection. Though self-collection might in some cases lead to contamination of the outer surface of the sample containers, the safe handling of such potentially contaminated containers by trained personnel in appropriately equipped central diagnostic laboratories poses only a minor infection risk. Moreover, since this simplified procedure has a better acceptability than NPS (4), it may lead to a higher willingness to submit to repeated

testing, accelerate the diagnostic process, and ultimately help avoid further spreading of the infection. In conclusion, this study highlights the usefulness of GL as an appropriate respiratory sampling method for symptomatic outpatients.

## ACKNOWLEDGMENTS

We thank Max Taubert for support with the statistical analyses.

J.Z., M.K., J.B., J.W.P.K., C.R.H.: conceived the study and performed experiments, analyzed data, prepared figures and drafted the manuscript; M.K., A.K., V.S.: analyzed samples and data, commented on the manuscript; C.Z., S.S., M.O.: recruited patients, procured samples, commented on the manuscript; J.Z., C.R.H.: wrote the manuscript. All authors approved the final version of the manuscript.

There was no funding for this work.

We declare no conflicts of interest.

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
