## [Reviewer comments · Microbiology Spectrum]

**Microbiology
Spectrum**

Self-collected gargle lavage allows reliable detection of SARS-CoV-2 in an outpatient setting

Johannes Zander, Stephan Scholtes, Maximilian Ottinger, Marcel Kremer, Azadeh Kharazi, Vanessa Stadler, Julia Bickmann, Christian Zeleny, Johannes Kuiper, and Christof Hauck

Corresponding Author(s): Christof Hauck, University of Konstanz

Review Timeline:

Submission Date:	June 10, 2021
Editorial Decision:	June 15, 2021
Revision Received:	June 18, 2021
Accepted:	June 21, 2021

Editor: Wendy Szymczak

Reviewer(s): The reviewers have opted to remain anonymous.

Transaction Report:

DOI: <https://doi.org/10.1128/Spectrum.00361-21>

June 15, 2021

Dr. Christof R Hauck
University of Konstanz
Lehrstuhl Zellbiologie
Universitaetsstrasse 10
Maildrop X908
Konstanz 78457
Germany

Re: Spectrum00361-21 (Self-collected gargle lavage allows reliable detection of SARS-CoV-2 in an outpatient setting)

Dear Dr. Christof R Hauck:

I have reviewed your responses to the previous reviewers, and I believe that your manuscript may be acceptable for Microbiology Spectrum if the following revisions are made:

1. Reviewer one points out that only two specimens in your study have a Ct value of >30 and suggest increasing the sample size. Your response indicated that your group feels that a sample size increase is not necessary since you have shown a statistical difference between high and low Ct specimens. I believe that the inclusion of more specimens would be difficult at this point in the pandemic and beyond the scope of the work; however, I would argue that the lack of specimens with high Ct values remains a limitation since the median value used to group high versus low Ct specimens (19.4) is very low. Please add a discussion of this limitation.
2. Line 64. Define or correct the abbreviation NLS

When submitting the revised version of your paper, please provide (1) point-by-point responses to the above as file type "Response to Reviewers," not in your cover letter, and (2) a PDF file that indicates the changes from the original submission (by highlighting or underlining the changes) as file type "Marked Up Manuscript - For Review Only". Please use this link to submit your revised manuscript - we strongly recommend that you submit your paper within the next 60 days or reach out to me. Detailed information on submitting your revised paper are below.

Link Not Available

Sincerely,

Wendy Szymczak

Journals Department
Reviewer comments:

Staff Comments:

Preparing Revision Guidelines

For complete guidelines on revision requirements, please see the Instructions to Authors at [link to page]. **Submissions of a paper that does not conform to Microbiology Spectrum guidelines will delay acceptance of your manuscript.**

Please return the manuscript within 60 days; if you cannot complete the modification within this time period, please contact me. If you do not wish to modify the manuscript and prefer to submit it to another journal, please notify me of your decision immediately so that the manuscript may be formally withdrawn from consideration by Microbiology Spectrum.

If you would like to submit an image for consideration as the Featured Image for an issue, please contact Spectrum staff.

Editorial Comments

1. Reviewer one points out that only two specimens in your study have a Ct value of >30 and suggest increasing the sample size. Your response indicated that your group feels that a sample size increase is not necessary since you have shown a statistical difference between high and low Ct specimens. I believe that the inclusion of more specimens would be difficult at this point in the pandemic and beyond the scope of the work; however, I would argue that the lack of specimens with high Ct values remains a limitation since the median value used to group high versus low Ct specimens (19.4) is very low. Please add a discussion of this limitation.

We also agree that in some settings or patient cohorts a larger portion of samples with Ct-values >30 (meaning with low viral titers) might occur. However, we would like to point out that we investigated mostly symptomatic persons, which had been tracked by health authorities as direct contacts of verified positive cases and which visited the doctor for the first time. This means that in contrast to studies on hospitalized COVID19 patients (which are usually days to even weeks after symptom onset) our study almost exclusively focussed on a patient cohort that is just before or right at the beginning of the symptomatic infection, when virus titers are highest (hence the median Ct value of 19.4 is rather low).

Of the 26 positive cases identified in our cohort, 4 (not 2) showed a Ct-value > 30. The original data for these four were:

Nasopharyngeal swab (NPS) – gargle lavage (GL)

30.1	33.0
30.2	32.9
32.4	34.6
34.6	34.5

As the four values labeled in bold are close to being identical, the data-points and also the connecting lines overlap almost completely in Fig. 1b. It is therefore almost impossible to separate these single data points by eye. We have tried to modify Fig. 1b using different symbols (open diamonds instead of rectangles), but it is still hard to resolve. Nevertheless, though this figure is not able to reveal every single data point, the main messages, that there is an overall shift to higher Ct-values, when using gargle lavage compared to nasopharyngeal swabs and that the difference is more prominent for the lower Ct-values, can be readily gained from this Figure.

We also explicitly discuss this observation, that Ct-values between NPS and GL show a more pronounced difference, when the Ct-values are smaller (hence, when the viral titers are high).

Furthermore, prompted by your suggestion, we now also highlight that in the few cases with Ct values >30 , the delta-Ct-values between NPS and GL are even smaller:

Mean delta-Ct of 8.8 for Ct-NPS values <19.4

Mean delta-Ct of 2.9 for Ct-NPS values ≥ 19.4

Mean delta-Ct of 1.9 for the 4 Ct-NPS values >30 Cq

To make these points (a small portion of samples (4) with high Ct-values, but also less of a difference between NLS and GL in these high Ct/low virus titer samples) better understandable for the reader, we have re-written the respective paragraph on page 5, line 11ff to now read:

“Overall, only a small portion ($n = 4$) of the positive samples showed a low viral titer with Cq values > 30 , in line with the idea that participants were identified early during the course of infection around symptom onset, when viral titers are highest (15). Interestingly, when we stratified the NPS samples into a lower half (Cq values below the median Cq value of 19.4) and into a higher half (Cq values equal or above 19.4) we observed that the differences in Cq values between NPS and GL samples were substantially more pronounced for samples, which showed lower Cq values in NPS (mean difference in Cq values between NPS and GL 8.8 for CqNPS values <19.4 ; $n =13$) (Figure 1C). In contrast, samples with higher Cq values in NPS (CqNPS values ≥ 19.4 ; $n =13$) had a reduced difference (mean difference of 2.9) in Cq values between NPS and GL (Figure 1C). Moreover, the 4 persons with Cq values in NPS > 30 showed an even smaller mean difference in Cq values between NPS and GL (1.9).”

2. Line 64. Define or correct the abbreviation NLS

We apologize for this mistake and replace “NLS” with “NPS”, as “nasopharyngeal swab” was meant.

June 21, 2021

Dr. Christof R Hauck
University of Konstanz
Lehrstuhl Zellbiologie
Universitaetsstrasse 10
Maildrop X908
Konstanz 78457
Germany

Re: Spectrum00361-21R1 (Self-collected gargle lavage allows reliable detection of SARS-CoV-2 in an outpatient setting)

Dear Dr. Christof R Hauck:

Your manuscript has been accepted, and I am forwarding it to the ASM Journals Department for publication. You will be notified when your proofs are ready to be viewed.

Sincerely,

Wendy Szymczak
Editor, Microbiology Spectrum
